# A Review on Canine and Human Soft Tissue Sarcomas: New Insights on Prognosis Factors and Treatment Measures

**DOI:** 10.3390/vetsci11080362

**Published:** 2024-08-10

**Authors:** Filippo Dell’Anno, Roberta Giugliano, Valeria Listorti, Elisabetta Razzuoli

**Affiliations:** 1National Reference Center of Veterinary and comparative Oncology (CEROVEC), Istituto Zooprofilattico Sperimentale del Piemonte, Liguria e Valle d’Aosta, 16129 Genova, Italy; filippo.dellanno@izsto.it (F.D.); valeria.listorti@izsto.it (V.L.); elisabetta.razzuoli@izsto.it (E.R.); 2Department of Public Health, Experimental and Forensic Medicine, Section of Biostatistics and Clinical Epidemiology, University of Pavia, 27100 Pavia, Italy

**Keywords:** soft tissue sarcoma (STS), comparative oncology, tumor micro-environment (TME), prognosis factor, immunotherapy

## Abstract

**Simple Summary:**

Soft tissue sarcomas (STSs) are rare tumors that develop from mesenchymal cells and can occur in both humans and dogs. Despite their rarity, STSs are challenging due to their tendency to recur and spread. Dogs represent valuable models for studying human STSs because they develop similar tumors naturally and share environmental risk factors with humans. Current treatments for STSs include surgery, radiation, and chemotherapy, but there is a need for new therapies due to the high failure rates of new drugs. In this study, new therapeutical approaches will be reviewed, since understanding the tumor microenvironment is crucial for developing better treatments, such as immunotherapy, for both dogs and humans.

**Abstract:**

Soft tissue sarcomas (STSs) represent a diverse group of tumors arising from mesenchymal cells, affecting both humans and animals, including dogs. Although STSs represent a class of rare tumors, especially in humans, they pose significant clinical challenges due to their potential for local recurrence and distant metastasis. Dogs, as a model for human STSs, offer several advantages, including exposure to similar environmental risk factors, genetic diversity among breeds, and the spontaneous development of tumors. Furthermore, canine tumors closely mimic the heterogeneity and complexity of human tumors, making them valuable for research into disease progression and treatment effectiveness. Current treatment approaches for STSs in both dogs and humans primarily involve surgery, radiation therapy, and chemotherapy, with treatment decisions based on tumor characteristics and patient factors. However, the development of novel therapeutic strategies is essential, given the high failure rate of new drugs in clinical trials. To better design new tailored treatments, comprehension of the tumor microenvironment (TME) is fundamental, since it plays a crucial role in STS initiation and progression by modulating tumor behavior, promoting angiogenesis, and suppressing immune responses. Notably, TME features include cancer-associated fibroblasts (CAFs), extracellular matrix (ECM) alterations, and tumor-associated macrophages (TAMs) that, depending on their polarization state, can affect immune responses and thus the patient’s prognosis. In this review, new therapeutical approaches based on immunotherapy will be deeply explored as potential treatment options for both dogs and humans with STSs. In conclusion, this review provides an overview of the current understanding of STSs in dogs and humans, emphasizing the importance of the TME and potential treatment strategies.

## 1. Introduction

Soft tissue sarcomas (STSs) are a heterogeneous group of tumors originating from mesenchymal cells and affecting humans and animals, including dogs. STSs represent a clinical challenge for local recurrence and distant metastasis. STSs can arise from any type of mesenchymal cell, including fibroblasts, adipocytes, smooth muscle cells, and skeletal muscle cells [1,2,3]. The most common subtypes of STSs in dogs include peripheral nerve sheath tumors (PNSTs), fibrosarcoma, myxosarcoma, liposarcoma, perivascular wall tumors, pleomorphic sarcoma (also termed malignant fibrous histiocytoma [MFH]), malignant mesenchymoma, and undifferentiated sarcoma [3]. Morphologically, both canine and human STSs can range from well-differentiated tumors to undifferentiated tumors with no recognizable tissue architecture [4,5]. Molecularly, in dogs and humans, STSs are characterized by complex chromosomal aberrations, gene mutations, the expression of oncogenes (MDM2–P53 and P16–CDK4–RB1), and the inactivation of tumor suppressor genes, along with alterations in signaling pathways, including the p53, *Rb*, and *PI3K/AKT/mTOR* pathways [6,7]. As reported by Das and Colleagues, RNA seq data on canine soft tissue sarcomsas demonstrated recurrent mutations occurring in *tumor protein* (*TP53)*, *histone lysine methyltransferase* (*KMT)* genes, and *platelet-derived growth factor B-chain gene PDGFB* fusions to be highlighted. Specifically, the authors found recurrent variants in the cancer-associated genes *KMT2D* and *TP53*, coupled with copy number losses of *retinoblastoma protein (RB1)* and *cyclin-dependent kinase inhibitor 2A (CDKN2A)*. A genomic characterization carried out on human soft tissue and bone sarcomas identified several frequently mutated genes in addition to previously known drivers such as KIT, SDHA, and PDGFRA. The newly identified genes include SETD2 and MAX, encoding, respectively, (i) a histone methyltransferase, a MYC binding partner, and a transcription factor, and ii) a MGA whose product binds the MAX-MYC complex [8]. In humans, the most common subtypes are liposarcoma, leiomyosarcoma, and undifferentiated pleomorphic sarcoma (previously called malignant fibrous histiocytoma) [9]. In dogs, STSs account for approximately 20% of malignant neoplasms of the skin [10,11]. A recent survey considering 70,966 histopathological diagnoses estimates that all sarcoma types account for around 13% of all canine tumors [12]. The incidence of STSs in dogs varies according to breed, age, and sex. Large-breed dogs, such as rottweilers, Doberman pinschers, and golden retrievers, are more prone to develop STSs than small-breed dogs [13,14]. Similarly, genome-wide association studies have correlated specific dog breeds (e.g., rottweilers, Rhodesian ridgeback, and mastiff) with the occurrences of sarcoma types (e.g., osteosarcoma) [15]. Most affected dogs tend to be middle-aged or older, with a median age reported between 10 and 11 years [16].

In humans, STSs represent less than 1% of all malignancies, with an incidence of 3–4 cases per 100,000 individuals per year [17]. Environmental pollution was suggested as a risk factor for sarcoma development in both dogs and humans, and several risk factors were identified, including exposure to ionizing radiation and chemicals [18,19]. In humans, occupational studies provided evidence of the role played by phenoxy herbicides, chlorophenols, and dioxins. The probability of developing sarcoma in subjects exposed to pesticides and dioxins increases by 85% and 156%, respectively, when compared to control cases. Even vinyl chloride exposure, the monomer of PVC (polyvinyl chloride) plastic manufacturing, was proven to play an important role in STS onset (risk ratio: 19.23, 95%; CI: from 2.03 to 182.46) [20].

However, the exact mechanisms through which environmental pollution increases the risk of STSs are not well understood, and further studies are necessary.

### Canine Model for Advancement in Soft Tissue Sarcoma Treatment

Dogs can be considered a valuable model for studying human STSs due to several factors. Firstly, dogs are exposed to environmental risk factors that are similar to their human companions, thereby making them an excellent model for environmental and lifestyle impacts on disease development [21]. Furthermore, distinct dog breeds have different susceptibilities to particular types of cancers. For instance, mast cell tumors are prevalent in golden retrievers, osteosarcoma are prevalent in greyhounds and, histiocytic sarcoma are prevalent in Bernese mountain dogs. These observations offer insights into the genetic basis of the disease. This can be beneficial for elucidating analogous genetic predispositions in humans [22]. Moreover, tumors in pet dogs occur spontaneously, which, in contrast to murine models, allows for the full representation of tumor heterogeneity and complexity. Similarly, pet dogs have an unmodified immune system, which allows researchers to fully evaluate the interactions between tumors and novel treatments [23]. Furthermore, dogs could be considered a suitable model for studying the progression of STSs and the effectiveness of treatments over a relatively short period, due to a shorter lifespan, more rapid disease progression, and a higher incidence rate. Both humans and dogs develop spontaneous cancers that share analogous characteristics with respect to tumor genetics. As reported by Rodrigues and colleagues, canine tumors share mutational hotspots with human tumors in oncogenes including *phosphatidylinositol-3-kinase* (*PIK3CA*), *Kirsten rat sarcoma virus* (*KRAS*), *neuroblastoma RAS viral oncogene* (*NRAS*), (*BRAF*) *serine/threonine-protein kinase B-Raf*, (*KIT*) *tyrosine-protein kinase KIT*, and (*EGFR*) *epidermal growth factor receptor* [24].

Despite considerable advancements in the characterization of STSs, the treatment of these tumors in both dogs and humans remains largely reliant on surgery, radiation therapy, and chemotherapy. The selection of an appropriate treatment depends on the size, location, and stage of the tumor, as well as the age and overall health of the patient. In dogs, complete surgical excision is the preferred treatment for STSs. However, adjuvant radiation therapy and chemotherapy may be necessary in cases of incomplete excision or metastatic disease [25]. The treatment of STSs in humans is more complex, necessitating a multidisciplinary approach. The main treatment approach is surgery with wide margins, but radiation therapy and chemotherapy may also be used in selected cases [26] (Figure 1).

Despite a 5-year disease-free survival rate exceeding 80% in human patients with localized soft tissue sarcomas, disease relapse occurs in over half of the cases, often with the development of distant metastasis [27]. Moreover, new approaches to cancer treatments are required to improve the current 85% failure rate of novel drugs in clinical trials [28,29,30]. The study of the tumor microenvironment (TME), defined as the internal and external environment in which tumor cells exist, including immune cells, stromal cells, blood vessels, and extracellular matrix (ECM), is facilitating the identification of new targets and consequently the development of new therapeutic protocols [31]. In both species, novel treatment options such as immunotherapy, targeted therapy, and gene therapy are being investigated, with promising results observed in preclinical studies [32,33,34,35] (Figure 1). 

Given the necessity of a more comprehensive understanding of the molecular mechanisms underlying STSs and of the development of novel and efficacious treatment options for this complex disease, the aim of this paper is to present an updated overview of TME and the therapeutic options available for the treatment of STSs in both dogs and humans.

## 2. Role of Tumor Microenvironment (TME) in the Onset and Progression of Canine and Human Soft Tissue Sarcomas

The tumor microenvironment (TME) plays a crucial role in the initiation, progression, and metastatic process by modulating tumor cell behavior, promoting angiogenesis and lymph angiogenesis, and suppressing immune surveillance [36]. 

### 2.1. Cancer-Associated Fibroblasts (CAFs), Extracellular Matrix (ECM), and Hypoxia

In general, the TME of STSs is a complex and dynamic network comprising tumor stromal cells, immune cells such as macrophages, lymphocytes, extracellular matrix (ECM), and signaling molecules that are capable of interacting and promoting tumor growth and invasion (Figure 2) [37,38,39]. 

Fibroblasts represent a major component of the TME and undergo activation by tumor cells, becoming cancer-associated fibroblasts (CAFs) [40]. The potential of CAFs as a target to treat STSs and sarcomas more generally is currently being investigated, given their capacity to activate proliferating mesenchymal cells during fetal development and malignancy [41,42]. Indeed, a study employing 3D co-culturing techniques strongly suggested the role played by canine CAFs in the activation, migration, and invasion of cancer cells [43]. CAFs secrete a range of factors, including growth factors, cytokines, and ECM components, which collectively favor tumor growth and metastasis by promoting cancer cells growth, enhancing pro-tumor immune responses, remodeling the ECM, influencing tumor cell drug resistance, and promoting angiogenesis [44,45]. As highlighted by Yuan and collaborators, the ECM exerts a profound influence on tumor development, affecting macromolecule components, degradation enzymes, and tissue stiffness [46]. ECM contributes to shaping an immunosuppressive environment, reducing the effectiveness of immunotherapies. For these reasons, the ECM was described as being capable of acting as a barrier to cancer treatments and supporting tumor progression.

Another typical feature of TME is hypoxia, which is the result of the uncontrolled proliferation of tumor cells, demanding additional nutrients, including oxygen. As suggested by Mortezaee and Majidpoor, in humans, hypoxia can result in a reduction in tumor immunogenicity and an increase in invasive clonal expansion of heterogeneous tumor cells. This is achieved by the weakening of the function of cytotoxic T lymphocytes (CTLs) and the attraction of regulatory T cells (Tregs) [47]. Moreover, a hypoxic environment promotes tumor progression by increasing the expression of several ECM remodeling enzymes such as LOX and collagen prolyl 4-hydroxylase (C-P4H) [48,49]. In view of the relevance of hypoxia in the development of STSs, Forker and colleagues validated a 24-gene signature as a biomarker of tumor hypoxia in STSs. With further validation, it could help in the selection of patients who: (i) exhibit a poor prognosis and an elevated risk of metastasis, ensuring their enrollment in clinical trials of neoadjuvant/adjuvant chemotherapy; (ii) may benefit from the addition of hypoxia modification via radiotherapy; and (iii) may be eligible for biomarker-driven trials of systemic hypoxia-targeted therapy [50].

### 2.2. Tumor-Associated Macrophages (TAMs) and Tumor-Infiltrating Lymphocites (TILs)

Another fundamental feature of TME is the presence of tumor-associated macrophages (TAMs), which are capable of inducing immunosuppression in sarcomas. In humans and dogs, TAMs can undergo two distinct polarizations, M1 or M2, starting from M0 populations, which are defined as unactivated macrophages [51]. M0 populations differentiate into the M1 subtype following stimulation with granulocyte–monocyte colony-stimulating factor (GM-CSF), lipopolysaccharide (LPS), or interferon-gamma (IFNγ). M1 populations promote a strong microbicidal and tumoricidal microenvironment through an enhanced antigen presentation capacity and the release of pro-inflammatory cytokines (IL1, IL-6, IL-12, or TNFa), reactive oxygen species (ROS), and nitric oxide (NO). Differently, M2 polarization, which is activated by M-CSF, IL-4, IL-10, and IL 13, induces a pro-tumoral immune suppressive environment through a high expression of PD-L1, IL-10, or TGF-β [52]. M2 TAMs can promote tumor growth and metastasis by stimulating angiogenesis, following the secretion of signal molecules and growth factors, including basic fibroblast growth factor (bFGF), platelet-derived growth factor (PDGF) and vascular endothelial growth factor (VEGF) [53]. A recent study conducted by Bray and colleagues on a population of 82 dogs indicated the possible use of VEGF and decorin, an angiogenesis inhibitor, as prognostic markers to predict tumor recurrence or dog survival in dogs affected by STSs [54]. The finding of the study showed, for the first time, that dogs diagnosed with STSs exhibiting high VEGF levels were 4-fold more likely to succumb to the disease than those with STSs displaying low VEGF levels [54]. Similarly, two studies analyzing human STSs reported a positive correlation between increased VEGF expression and a higher tumor grade. However, the authors were unable to confirm an association with the clinical outcomes due to insufficient data [55,56]. Such observations support the validity of the canine model for the study of the prognostic factors of STSs in the field of comparative oncology. Moreover, studies on human sarcomas have demonstrated that TAMs exert a negative regulatory effect on cytotoxic effector cells, such as CD8+, natural killer (NK), and NKT cells. This is due to their frequent expression of programmed death-ligand 1 (PD-L1)/L2, B7-H4, and V-domain Ig suppressor of T cell activation (VISTA), which trigger the inhibitory PD-1-mediated immune checkpoint in T cells [57]. Additionally, TAMs promote the expansion of immunosuppressive Tregs and myeloid-derived suppressor cells (MDSC) through the interaction of cytokines and metabolic enzymes with surface receptors. Indeed, Tregs inhibit the function of effector T cells and (NK) cells [58], while MDSCs suppress the function of immune cells by producing 1 (arginase 1 (ARG1)), which in turn favors tumor cell growth and inhibits immune cell function [59]. Notably, activated T cells and Treg subsets can express cytotoxic T lymphocyte-associated protein 4 (CTLA-4, CD152) on their cell surface. This can be used as an immune checkpoint molecule, resulting in the down-regulation of T cells and the inhibition of the antitumor response [60].

It is generally accepted that a higher abundance of tumor-infiltrating lymphocytes (TILs) compared to tumor-associated macrophages (TAMs) in the TME is associated with a better prognosis for patients affected by STSs [61]. Recently, Watanabe and colleagues demonstrated that human patients with advanced soft tissue sarcomas and a high lymphocyte/monocyte ratio (LMR) (>2) had a better prognosis than patients with a low LMR. Moreover, the LMR correlates with the tumoral CD3/CD68 ratio, which is a marker of the T cell and monocyte lineage [62]. Such data need to be confirmed by further experiments, as the prognostic role of cellular markers may vary considerably between different tumors in terms of predictability. For example, the prognostic value of the marker CD163+ (associated with M2 macrophages) differs among different diseases in humans. As reported by Pillozzi and colleagues, low CD163+ levels are associated with favorable survival in synovial sarcoma, while high levels of CD163+ are associated with enhanced survival in embryonic rhabdomyosarcoma [17]. Similarly, a study conducted on dogs affected by osteosarcoma pointed out that high CD204 expression, commonly associated with an M2 phenotype, was associated with an improved survival [63]. Although osteosarcomas are not considered STSs, they share a mesenchymal origin and a propensity for localizing primarily in the extremities, with less-frequent occurrences in the retroperitoneum and trunk. Recent studies demonstrated that comparative analyses can provide useful data for understanding and developing potential therapies for both conditions [63,64,65].

Overall, to better clarify still-controversial processes and to explore the viability of such cellular markers as effective tools in comparative oncology, the development of technologies based on next-generation sequencing is allowing further progress in the identification of key interactors in the onset and development of sarcomas. 

### 2.3. Novel Prognostic Factors and Molecular Markers Associated with Canine and Human STSs

Prognostic factors combined with molecular markers are of fundamental importance to determine the most appropriate STS therapy. Histologically, a high mitotic index (number of mitoses per n° high power fields (HPFs) or number of mitoses per 2.37 mm^2^) is associated with a reduced survival time in both humans and dogs [3,66]. Technological progress offers the possibility of identifying early prognostic and predictive factors, thus favoring timely and targeted therapies.

### 2.4. Genes as Predictors of STSs

Advancements in genetic and molecular profiling allowed several abnormalities located in the *phosphatidylinositol-4,5-bisphosphate 3 kinase catalytic subunit alpha (PIK3CA)* and *neuroblastoma RAS viral oncogene homolog (NRAS)* pathways to be identified in canine sarcomas. Mutations in tumor protein p53 (*TP53*), *phosphatase and tensin homolog* (*PTEN*), and *cyclin dependent kinase inhibitor 2A* (*CDKN2A*) were also found [67]. A recent study conducted on the transcriptome of 71 dogs with osteosarcoma metastases revealed the presence of two transcript clusters differing from each other due to the different numbers of metastatic processes observed [68]. The analysis showed that patients belonging to Cluster 1 were had significantly enriched pathways involved in interferon alpha response, while patients in Cluster 2 had significantly enriched pathways such as *G2M* checkpoint and *E2F* target genes and inflammatory response pathways, and the pathways for *KRAS* signaling and myogenesis were downregulated. Since dogs in Cluster 1 displayed fewer metastases and a complete absence of lung metastases, these data suggest that this type of immune response pathway may somehow be protective for metastatic colonization of tissues or organs [68]. In human STSs, a recent work on tumor transcriptomes demonstrated the crosstalk between RNA modification regulators and the potential roles in TME and immune infiltrates. In detail, the authors identified several RNA regulators associated with the onset of the disease. The expression levels of *methyltransferase 14* (*METTL14*), *Wilms Tumor 1-Associating Protein* (*WTAP*), *YTH N6-Methyladenosine RNA Binding Protein C1*(*YTHDC1*), and *Leucine Rich Pentatricopeptide Repeat Containing* (*LRPPRC*) were significantly lower in STS cell lines compared with the expression of human skin fibroblasts, while *METTL3*, *METTL16*, and *insulin-like growth factor binding proteins 2* (*IGF2BP2)* were mainly associated with tumor cells [69]. 

### 2.5. RNAs as Possible Diagnostic Factors

Han and colleagues confirmed the role of long noncoding RNA in cancer progression and development. The authors showed that the cuproptosis-associated lncRNAs can effectively predict the tumor immune microenvironment in STS patients, and that high-risk patients are more likely to have immunosensitive tumors that are more responsive to immunotherapy [70]. Similarly, microRNAs were proposed as possible diagnostic and/or prognostic markers. Ye and colleagues identified 57 differentially expressed microRNAs following high-throughput sequencing of serum exosomes from human osteosarcoma patients; the analysis revealed that 20 microRNA were upregulated and 37 were downregulated. The authors found an increased expression of exosomal *miR-195-3p*, which could promote the proliferation and invasion of sarcoma cells [71]. Additionally, the serum levels of exosomal *miR-1260b*, which is associated with tumor burden and with the infiltrative ability of myxofibrosarcoma, were found to be overexpressed in ten preoperative samples of patients with infiltrative myxofbrosarcoma [72]. Following single-cell transcriptome analysis, Qi and colleagues identified genes that are differently expressed in the anoikis process occurring in human STSs [73]. Notably, anoikis is a programmed cell death mechanism that is activated in non-tumoral tissues when a cell is detached from its original extracellular matrix. The authors proposed the use of anoikis gene expression patterns as a prognostic factor, as patients with enriched immune-related pathways such as *IL6 JAK-STAT3* signaling, *TNFA* signaling, complement, *INFγ* response, and INFα response showed a better prognosis [73].

Although the studies reported so far are of fundamental importance in advancing the diagnosis, prognosis, and treatment of soft tissue sarcomas, most of the information is deriveed from studies conducted in humans rather than in dogs. For the reasons outlined in the introduction to this (mini)review, an increase in the number of in-depth analyses of the TME of STSs in the canine model could accelerate our knowledge of this type of disease, with a consequent improvement in the quality of life of both humans and dogs. In this regard, Bray and Munday developed a nomogram; a statistical tool capable of calculating an objective risk measure by analyzing different patient and tumor characteristics related to TME, allowing the clinician to decide whether adjuvant therapy or further surgery should be considered to reduce the risk of recurrence after surgery for STSs in the canine model [74]. Even though the nomogram can be further improved in terms of sensitivity and specificity, its authors were able to accurately predict tumor-free survival in 25 canine patients, paving the way for translating such tools into human medicine.

## 3. New Therapies for Canines and Human Soft Tissue Sarcoma

Soft tissue sarcomas (STSs) are common cutaneous or subcutaneous neoplasms in dogs. Many STSs are initially treated through surgical excision. Although many STSs can be successfully treated using this approach, local recurrence can occur in around 20% (range of 7–75%) of the patients, reducing overall survival in dogs [3,75]. The standard surgical treatment for canine STSs involves the excision of the neoplastic mass with wide margins (a minimum of 3 cm for lateral margins and one clean fascial plane for deep margins is recommended [76]). However, as suggested by Abrams et al., further studies are needed to provide a universal system of margin reporting to better address the treatment of STSs in dogs [77]. Indeed, the degree of resection and completeness of the surgical margins are important prognostic factors. Moreover, only 7% of low-grade tumors recurred after margin excision, compared to a 75% recurrence rate observed for intermediate and high-grade tumors [16,73]. To overcome the above-listed issues, treatment protocols combining surgery with adjuvant therapies such as radiotherapy, chemotherapy, and immunotherapy, alone or in combination, have been proposed in both humans and dogs (Table 1a–d). 

Radiotherapy may be employed either before (neoadjuvant) or after (adjuvant) surgery. In humans, radiotherapy combined with surgery has proven to be effective in decreasing the needed of surgical margins without compromising patient outcomes [88]. In general, neoadjuvant radiotherapy is widely used to treat human STSs, except for low-grade STS and for peculiar tumor locations. The National Comprehensive Cancer Network (NCCN) [115] guidelines recommend either preoperative or postoperative radiation for stage II, IIIA, and IIIB extremity STSs. Surgery alone may be considered for stage IA or IB tumors resected with wide margins. Conversely, chemotherapy is often limited in STSs in adults; as such, tumors rarely show high chemosensitive histological profiles, which is more common in pediatric settings. According to van der Graaf and colleagues [116], pediatric sarcomas, including embryonal and alveolar rhabdomyosarcoma, Ewing’s sarcoma, and osteosarcoma, are indeed characterized by an increased chemosensitivity if compared to other sarcomas such as synovial sarcomas. As reported by Dantonello et al., following a multi-institutional data review, the data suggested that approximately 90% of clinical group III embryonal rhabdomyosarcoma (RMS) patients experienced a volume reduction of 33% or greater after induction chemotherapy. While the impact of chemotherapeutic response on event-free survival (EFS) is unclear, the research suggests that individuals with at least a partial response experienced a significant improvement in overall survival (OS), especially in those with head and neck RMS [117]. 

As suggested by Linch and colleagues and Gamboa et al. [118,119], histology and grade as well as tumor location are fundamental to establish which therapy may be the most effective to treat human STSs. Most localized extremity STSs are best treated surgically with or without radiotherapy. However, the results reported by Gamboa et al. show that patients with high-grade extremity STSs measuring >10 cm obtain benefits from neoadjuvant chemotherapy. More generally, chemotherapy is used to treat patients who show metastatic disease, either initially or after removal of the primary tumor. It is also occasionally used to help shrink very extensive tumors that may not be suitable for limb-saving surgery [120]. 

In the canine model, treatments of STSs relying on radiotherapy and chemotherapy represent an option for tumors that are too large to resect or inaccessible and for grade III tumors [121,122,123]. Recently, a combination of radiotherapy and immunotherapy, an approach based on the stimulation of the immune system to recognize and attack cancer cells, has been proposed for the treatment of spontaneous tumors, including STSs, in dogs. A study by Boss and colleagues evaluated the combined use of stereotactic body radiotherapy (SBRT) and of local injection of immunomodulatory antibodies to OX40 and TLR 3/9 agonist for the treatment of STS in dogs [90]. The authors evaluated the efficacy of SBRT alone or in combination with immunotherapy. Two weeks after treatment with SBRT alone, the results showed increased tumor densities of CD3+ T cells, FoxP3+ Tregs, and CD204+ macrophages, along with increased expression of genes associated with immunosuppression. The addition of OX40/TLR3/9 immunotherapy to SBRT resulted in local depletion of Tregs and tumor macrophages and reduced Treg-associated gene expression (FoxP3), suppressed macrophage-associated gene expression (IL-8), and suppressed exhausted T cell-associated gene expression (CTLA4) [90]. Although the initial aim of the authors was not to assess therapeutic efficacy, but to simply describe post-treatment modifications in TME, these early findings provide the rationale for further follow-up studies.

Approaches based on classical adjuvant chemotherapy relying on doxorubicin administration did not show promising results, since in some cases, an absence of response, or at best, little benefit to patient outcomes, were observed [96,97]. In contrast, a recent study on dogs with STSs reported tumor remission in the majority of treated patients 60 days after a combined treatment of high-intensity focused ultrasound (HIFU) and doxorubicin [95]. Another study conducted on canine STSs based on the use of chemotherapeutics such as thalidomide, piroxicam, and cyclophosphamide did not improve the progression-free interval or the overall survival [93]. Torrigiani and colleagues proposed canine STS treatment based on electrochemotherapy (ECT), a technique for the treatment of locally invasive cutaneous primary and secondary tumors using electric pulses to enhance the transmembrane delivery of cytotoxic drugs such as bleomycin and cisplatin [102]. This approach has shown promise, with treatment resulting in two complete remissions and one partial remission. Notably, post-ECT toxicity was recorded to be mild in 66.7% of cases [102]. These results confirmed the observations of Campana and colleagues [103]. The authors observed a response rate of 85% after ECT combined with bleomycin in human STSs. Moreover, electroporated tumors regressed completely in one-third of the patients after a single treatment. Unfortunately, adverse effects such as skin ulcerations or soft tissue necrosis were observed in 35% of the patients. Further research is needed to refine these alternative treatments and optimize their efficacy and safety profiles.

### Immunotherapy Strategies for Canine STS Treatments

Immunotherapy has emerged as a promising treatment modality for different cancers (e.g., acute lymphoblastic leukemia, B-cell non-Hodgkin lymphoma, breast cancer, etc.) including STSs in both humans and dogs. Immunotherapy can be defined as a medical approach that harnesses the body’s own immune system to treat diseases, particularly cancer and autoimmune disorders. It is designed to stimulate or enhance the body’s natural immune response, making it more effective in identifying and eliminating harmful cancer cells or those causing autoimmune reactions [124]. There are different types of immunotherapies available to treat STSs, depending on the target. In general, immunotherapy involves different approaches, including: (i) checkpoint inhibitors, which are able to inhibit proteins that prevent immune cells from attacking cancer cells, (ii) monoclonal antibodies, which are designed to target specific proteins on the surface of cancer cells, allowing the immune system to recognize and attack them more effectively, (iii) adoptive cell therapy and CAR-T cell therapy, both based on the modifications of the patient’s own immune cells to better recognize and attack cancer cells, (iv) cytokines, proteins capable of boosting the immune system’s response, and (v) vaccines that can stimulate the immune system to produce an immune response against cancer-specific antigens [124,125]. Recent studies have investigated the use of immunotherapeutic agents, alone or in combination with other treatments (e.g., radiation therapy, cytotoxic chemotherapy), for managing STSs in dogs [34,126] (Figure 3). 

Among the immune checkpoint inhibitor therapies, those targeting the interactions between the immunosuppressive receptor expressed on the surface of T cells, namely Programmed Death-1 (PD-1) and its ligands PD-L1 and PD-L2, are considered promising, especially in tumors with high PD-1 expression and tumor-infiltrating lymphocytes (TILs) [127]. PD-L1, which is frequently expressed on the surface of malignant tumor cells, induces their apoptosis after binding to PD-1 on T cells, thereby promoting immunotolerance, tumor progression, and metastasis [128]. In humans, the efficacy of two anti-PD-1 antibodies, pembrolizumab and nivolumab, was demonstrated after the treatment of patients with undifferentiated pleomorphic sarcoma and metastatic sarcoma, respectively [105,106]. The application of such an approach in dogs is limited, since little information is available in the literature about PD-1/PD-L1 expression. Recently, Stevenson and colleagues reported the first characterization of checkpoint molecules’ expression in canine STSs [129]. Their results highlighted a statistically significant increase in the expression of PD-1 in grade 3 STSs compared to grade 1, while the expression of PD-L1 was found to be higher in grade 2 STSs compared with grade 1 STSs. The expression of PD-L2 increased with tumor grade, although it did not reach statistical significance. Recently, mouse anti-canine PD-L1 monoclonal antibodies have been successfully tested on formalin-fixed paraffin-embedded tissue from canine cancers, including sarcomas [130]. Given their promising capabilities as therapeutic agents in dogs and their potential application as prognostic markers, such antibodies require further investigation.

An interesting insight into the treatment of canine STs was provided by Stinson and colleagues, who reported the efficacy of interleukin 2 and interleukin 12 [104]. To enhance the therapeutic window of such compounds, the cytokines were engineered to bind and anchor to tumor collagen following inter-tumor injection. Enhanced T cell infiltrates and an increase in gene expression associated with cytotoxic immune function were shown [104].

Another interesting approach to enhance antitumor immunity is the use of anaerobic bacteria. A genetically engineered *Salmonella enterica* serovar Typhimurium was evaluated against canine osteosarcoma [107,108]. Reduced toxicity and a prolonged disease-free interval (DFI) were observed compared with the control group. *Salmonella* was not observed in any of the tissues sampled, suggesting that the beneficial effects were mainly due to an antitumor immune response rather than direct tumoricidal activity of the vector. Indeed, lymphocytosis and monocytosis were observed in 18 out of 19 dogs 10 days after the first *Salmonella* administration. A recent study by Razzuoli and colleagues reported promising results following exposure of wild-type *Salmonella typhimurium* to the canine STS cell line A-72 [109]. Indeed, *S. typhimurium* invaded A-72 cells, inducing a pro-inflammatory response associated with a decrease in cell viability. After treatment, increased expression of CXCL8, NOS2, CXCR4, and PTEN was observed, while TP53 was slightly expressed, as shown in human STSs. Such results suggest the possibility of using a bacterial agent to stimulate the expression of important genes involved in the innate immune response. Even though further trials are needed, the study also provides a useful model for the in vitro evaluation of new therapeutic approaches that could be translated into human oncology [107]. Interest in oncolytic virus-based therapies is also growing in the scientific community. Beguin and collaborators reported the possibility of using TG6002, an oncolytic vaccinia virus expressing the FCU1 protein, which converts 5-fluorocytosine to 5-fluorouracil (a chemotherapeutic compound) to treat various canine tumors, including six patients with STSs [111]. According to the authors, effective viral replication, targeted intra-tumoral 5-FU production, and reversion of the immunosuppressive TME due to the immune response associated with TG6002 and 5-FC administration were confirmed. Le Beouf and colleagues [114] highlighted the appropriateness of the canine model by reporting the successful use of the oncolytic Maraba virus *(MG1*) to treat in vitro lines of human and canine sarcomas. Similarly, an attenuated MYXV virus was assessed as a possible therapeutic agent in canine STSs [113]. The authors injected the virus, coupled with oclacitinib, decreasing type I IFN and thus increasing MYXV replication in canine cancer cells. The injection was performed around the surgical site at two time points post-operatively to treat the residual tumor cells. The treatment did not appear to affect the rate of tumor recurrence in dogs, but it provided interesting insights into the combined use of oclacitinib, since it effectively increased MYXV replication. 

The difficulty in interpreting and comparing clinical outcomes between the two models is due to several underlying factors. As highlighted by Choi and colleagues [131], the limited understanding of canine immunobiology is a significant challenge. Unlike in humans, where lymphocyte populations and immune checkpoint pathways have been extensively characterized, this knowledge remains comparatively rudimentary in dogs. This knowledge gap limits the direct translation of findings and hinders the ability to draw clear parallels between the two models. Moreover, the availability of essential tools for immune profiling and therapeutic targeting in canine oncology is limited. Commercially available staining and blocking reagents tailored to specific immune checkpoint pathways, which are widely used in human oncology, are not consistently accessible for canine studies. Given the extensive range of reagents used in human immunotherapy, the absence of analogous resources for dogs demands for the development and validation of additional reagents to facilitate the establishment of effective therapeutic strategies in the canine model. Addressing these challenges demands a concerted effort to advance our understanding of canine immunology and to develop the tailored resources that are essential for elucidating and exploiting therapeutic avenues in veterinary oncology.

## 4. Conclusions

Recent studies have outlined the importance of CAFs in TME. It is known that CAFs can produce ECM components and compounds that promote tumor growth and metastasis. ECM strongly influences tumor development and contributes to an immunosuppressive microenvironment, thus reducing the efficacy of immunotherapies and chemotherapy. Based on the reviewed literature, complete surgical excision is the treatment of choice for canine STSs, while adjuvant radiotherapy and chemotherapy may be necessary in cases of incomplete excision or metastases. In humans, the treatment of STSs is more complex; however, also in humans, wide-margin surgery is the mainstay of the treatment, followed by radiotherapy or chemotherapy in selected cases. One promising approach to STS therapy is immunotherapy and its ability to directly stimulate the patient’s immune response. Overall, as most immunotherapy studies involve the human model, more research is needed in canine STSs. The scarcity of information on the TME and therapies for canine STSs is the most limiting factor for a systematic and conclusive description of the similarities and differences between the two models. Finally, future studies on the different aspects of STSs will lead to a better comprehension of the disease and thus its treatment.

## Figures and Tables

**Figure 1 vetsci-11-00362-f001:**
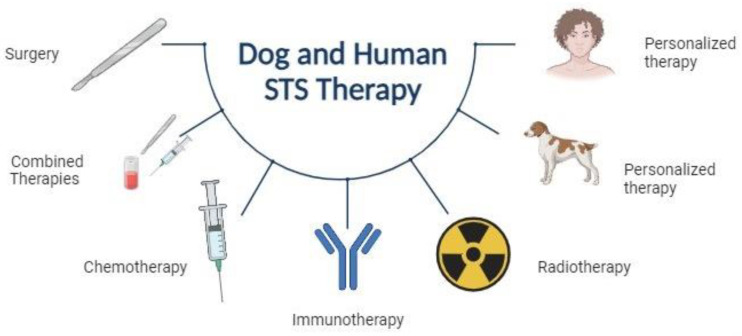
A resume of the available therapies for humans and dogs in the treatment of soft tissue sarcomas.

**Figure 2 vetsci-11-00362-f002:**
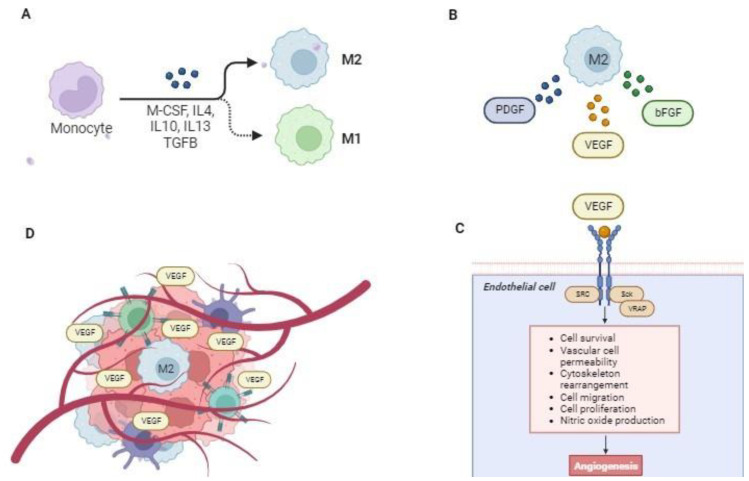
Tumor microenvironment in human and canine soft tissue sarcomas. (**A**) Tumor-associated macrophages (TAMs) can undergo two different polarizations, M1 or M2, starting from M0 populations defined as non-activated macrophages. M0 populations differentiate into the M1 subtype after stimulation with granulocyte–monocyte colony-stimulating factor (GM-CSF), lipopolysaccharide (LPS), or interferon-gamma (IFNγ). M2 polarization is activated by M-CSF, IL-4, IL 10, and IL 13, and it induces a pro-tumoral environment. (**B**) M2 can release signaling molecules and growth factors such as basic fibroblast growth factor (bFGF), platelet-derived growth factor (PDGF), and vascular endothelial growth factor (VEGF). (**C**) The activation of the VEGF pathway is responsible for angiogenesis and tumor growth. (**D**) VEGF can be used as a prognostic marker to predict tumor recurrence or survival in STSs. Indeed, there is a positive correlation between increased VEGF expression and a higher tumor grade. Moreover, TAMs negatively regulate cytotoxic effector cells, such as CD8+ and natural killer (NK) cells, and they often express programmed death-ligand 1 (PD-L1)/L2.

**Figure 3 vetsci-11-00362-f003:**
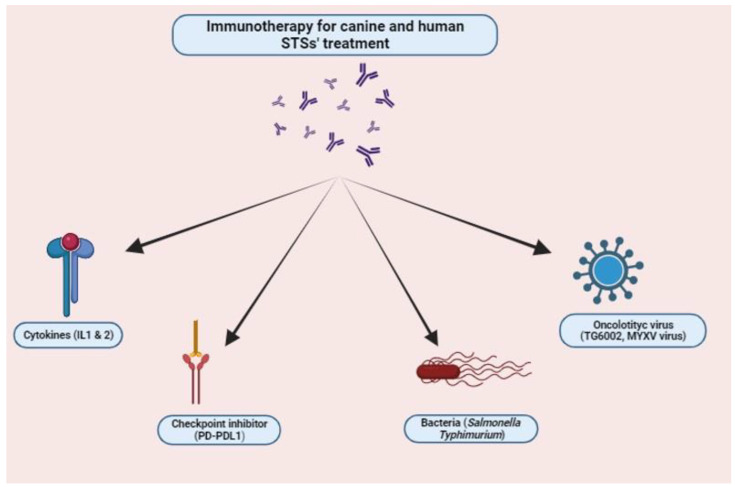
Immunotherapy approaches specific to treat canine and human soft tissue sarcomas. See Table 1 for a more detailed description.

**Table 1 vetsci-11-00362-t001:** **a**: Surgical treatments in dog and human STSs. **b**: Ablation treatments in dog and human STSs. **c**: Combined treatments in dog and human STSs. **d**: Immunotherapy treatments in dog and human STSs.

**a**					
**Dog**			**Human**		
**Treatment Specification**	**Reference**	**N° of Cases**	**Treatment Specification**	**Reference**	**N° of Cases**
**Standard Surgery**					
Excision with margins of 2 to 3 cm is considered to be the optimal goal when surgery is the sole treatment modality.	[3,16,54,74,78]	97 350	Quality of surgical margins independently predicted local control and survival.	[79,80]	997 197
**b**					
**Dog**			**Human**		
**Treatment specification**	**Reference**	**N° of Cases**	**Treatment Specification**	**Reference**	**N° of Cases**
**Percutaneous Image-Guided Treatments with Ablation Technologies (i.e., Radiofrequency Ablation, Cryotherapy, Microwave Ablation)**					
To treat superficial, small, non-invasive lesions, or when definitive surgery cannot be performed because of limitations imposed by regional anatomy or owner reluctance	[81,82,83]	20 42 **6**	Compared with surgery, image-guided treatments provide a lower-morbidity option with excellent tolerance and preservation of long-term function with low damage to healthy parenchyma	[84,85,86]	2145 65 1
**c**					
**Dog**			**Human**		
**Treatment specification**	**Reference**	**N° of cases**	**Treatment specification**	**Reference**	**N° of cases**
**Radiotherapy Combined with Surgery**					
First-in-dog clinical trial on NK cells (CD5^dim^, NKp46+) isolated from PBMCs and expanded with irradiated K562-C9-mIL21 feeder cells and recombinant human IL-2	[87]	13	Important role in limb preservation during treatment for extremity soft tissue sarcomas. Preoperative radiotherapy offers the potential for lower radiotherapy doses but likely increases acute toxicity for thigh tumors	[88,89]	96 20
**Radiotherapy Combined with Immunotherapy Treatment**					
In situ vaccination approach combined an OX40 agonist antibody with TLR 3 and TLR 9 agonists	[90]	997	Nivolumab and ipilimumab are monoclonal antibodies targeting PD-1 and CTLA-4, respectively	[91]	38
In situ vaccination was subsequently combined with targeted radionuclide therapy using a theranostic pairing of IV 86Y-NM600 and IV 90Y-NM600	[92]	17	Olaparib + radiotherapy using Poly-ADP ribose polymerase (PARPi)	[91]	38
**Radiotherapy Combined with Chemotherapy**					
Metronomic chemotherapy using thalidomide, piroxicam, cyclophosphamide	[93]	50	3D conformal radiotherapy combined with ifosfamide and doxorubicin administration	[94]	115
**Combined Chemotherapy**					
Combined with high-intensity focused ultrasound (HIFU) for treating solid tumors using thermal and histotripsy-based mechanical ablation	[95]	1	Doxorubicin, ifosfamide, and lenograstim administration	[96,97]	351 1953
Combined with surgery shows a significant difference in survival times compared to dogs treated with surgery alone	[98]	23	Metronomic chemotherapy	[99]	45
Maximum-tolerated-dose chemotherapy vs. metronomic chemotherapy	[100]	103	In a subset of patients with advanced leiomyosarcoma, the authors administered temozolomide with concomitant thalidomide	[101]	33
**Electrochemotherapy (ECT)**					
Antitumor local ablative treatment that uses electric pulses to enhance the intracellular delivery of bleomycin	[102]	52	Antitumor local ablative treatment that uses electric pulses to enhance the intracellular delivery of bleomycin	[103]	34
**d**					
**Dog**			**Human**		
**Treatment Specification**	**Reference**	**N° of Cases**	**Treatment Specification**	**Reference**	**N° of Cases**
**Immunotherapy**					
Authors engineered (i.e., caninized) cytokines that bind and anchor to tumor collagen	[104]	10	Pembrolizumab and nivolumab administration	[105,106]	86 96
**Immunotherapy Using Anaerobic Bacteria**					
Using *Salmonella enterica* serovar Typhimurium.	[107,108,109]	23 351	Using *Salmonella enterica* serovar Typhimurium	[110]	n.a.
**Immunotherapy Using Oncolytic Virus**					
Vaccinia virus TG6002 and 5-fluorocytosine administration	[111]	13	Oncolytic virus JX-594 combined with cyclophosphamide	[112]	20
Myxoma virus inoculation	[113]	103			
Maraba virus MG1: in vivo test	[114]	10	Maraba virus MG1: in vitro test	[114]	10

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
