# Peer review of "A Review on Canine and Human Soft Tissue Sarcomas: New Insights on Prognosis Factors and Treatment Measures"

_vetsci, 2024, doi:10.3390/vetsci11080362_

Round 1

Reviewer 1 Report

Comments and Suggestions for Authors

This manuscript is a review of the current literature with respect to canine and human soft tissue sarcomas (STS), with a focus on new information, and the possible relationship of that to treatment. As such, it does not exhaustively cover many factors associate with STSs. While this is understood, there could be more information provided as background to further substantiate the similarities (and differences) in these tumors between dogs and humans.

Specific issues noted include:

Line 68 contradicts line 52-53 with respect to the frequency of lipo- and leiomyosarcoma.

Line 90-91 – perhaps expound upon how these confounders affect onset? What are the effects of age and sex?

The authors use osteosarcoma as an example several times, but it doesn’t fit the topic of soft tissue sarcoma. 

The authors use the word “various” (as well as other words) to modify a term, such as oncogenes, in multiple places. This use is not informative as it does not specify which oncogenes, which would be much more informative. I suggest searching the document for “various”, “certain”, and any other occurrences of this failure to list details and changing them to actually list the factors involved. 

Comments on the Quality of English Language

The manuscript needs significant English language editing. There are numerous examples throughout the manuscript of issues with tense, case, sentence structure, and phrasing. 

Examples of needed edits (I stopped after line 115, but these are numerous throughout the document):

Line 16 – delete both commas

Line 29 – delete “the” in front of comprehension

Line 58-59 – “…allowed to highlight recurrent mutations occurring on…” is poorly worded. Suggest rewording to “…allowed recurrent mutations occurring in…(list of tumors)… to be highlighted.”

Line 66 – delete “for”

The single sentence paragraph structure in line 79-80, and again in 81-83 seems a bit odd. Perhaps these could be combined into a single larger paragraph about the human tumors?

Line 86-87  - might be better reworded to “…exposed to pesticides was 85%, while dioxin exposure lead to an increase of 156% when compared to control cases.”

Line 90 – might be better reworded to “…onset of STSs…”

Line 104 – insert “dogs” so that it reads “…allows dogs to fully…”

Line 109 – perhaps change “a quicker diseases progress and a higher number of per year STSs diagnosis” to “more rapid disease progression and a higher incidence rate”.

Line 114 – the gene names for (BRAF), (KIT) and (EGFR) were not provided. 

Line 115 – “allow to” should be “allowed the authors to”

Author Response

This manuscript is a review of the current literature with respect to canine and human soft tissue sarcomas (STS), with a focus on new information, and the possible relationship of that to treatment. As such, it does not exhaustively cover many factors associate with STSs. While this is understood, there could be more information provided as background to further substantiate the similarities (and differences) in these tumors between dogs and humans.

A:  We thank the reviewer for suggestions that improve the quality of the manuscript. However, we ask the reviewer, if possible, to indicate further works that can be cited in the text to better describe the background relating to STS in dogs and human counterparts. Indeed, we found a small number of papers on canine STS useful for a comparative analysis with humans.

Specific issues noted include:

Line 68 contradicts line 52-53 with respect to the frequency of lipo- and leiomyosarcoma.

A: Tumors have different frequencies in the two models, but it is a fact reported in the literature.

Line 90-91 – perhaps expound upon how these confounders affect onset? What are the effects of age and sex?

A: The effect of age and sex cannot be known. The authors themselves report the following sentence:” Many studies excluded female workers due to small numbers and rarity of involvement in highly exposed occupations. Therefore, subgroup analysis based on gender was not performed due to population selection bias.”

The authors use osteosarcoma as an example several times, but it doesn’t fit the topic of soft tissue sarcoma. 

A:  The reviewer is right. However, given the paucity of information on canine STSs we used information relating to osteosarcoma to provide, by comparison, useful information relating to STSs. To this end, we have nevertheless stated the difference between osteosarcoma and STSs. Moreover, we have cited studies in the literature that highlight the usefulness of comparing the two diseases (line 256)

The authors use the word “various” (as well as other words) to modify a term, such as oncogenes, in multiple places. This use is not informative as it does not specify which oncogenes, which would be much more informative. I suggest searching the document for “various”, “certain”, and any other occurrences of this failure to list details and changing them to actually list the factors involved. 

A: According to the reviewer suggestion we deleted or modify the word “various” and/ or certain along the text. Amendements can be found at line: 56, 76, 102, 178, 282, 428, 431, 443, 530.

Comments on the Quality of English Language

The manuscript needs significant English language editing. There are numerous examples throughout the manuscript of issues with tense, case, sentence structure, and phrasing. 

Examples of needed edits (I stopped after line 115, but these are numerous throughout the document):

Line 16 – delete both commas

A: Done

Line 29 – delete “the” in front of comprehension

A: Done

Line 58-59 – “…allowed to highlight recurrent mutations occurring on…” is poorly worded. Suggest rewording to “…allowed recurrent mutations occurring in (list of tumors)… to be highlighted.”

A: amended as required

Line 66 – delete “for”

 A: amended as required

The single sentence paragraph structure in line 79-80, and again in 81-83 seems a bit odd. Perhaps these could be combined into a single larger paragraph about the human tumors?

A: As suggested we created a unique paragraph about the human tumors.

Line 86-87  - might be better reworded to “…exposed to pesticides was 85%, while dioxin exposure lead to an increase of 156% when compared to control cases.”

A: amended as required

Line 90 – might be better reworded to “…onset of STSs…”

A: amended as required

Line 104 – insert “dogs” so that it reads “…allows dogs to fully…”

 A: amended as required

Line 109 – perhaps change “a quicker diseases progress and a higher number of per year STSs diagnosis” to “more rapid disease progression and a higher incidence rate”.

A: amended as required

Line 114 – the gene names for (BRAF), (KIT) and (EGFR) were not provided. 

A: amended as required

Line 115 – “allow to” should be “allowed the authors to”.

A: amended as required

As suggested, all the manuscript has been checked by a native English speaker.

Reviewer 2 Report

Comments and Suggestions for Authors

Thank you to the authors for submitting this work.  While it is very interesting and well researched for some aspects, my largest concern is that the discussion includes histologic types of tumors not typically classified as soft tissue sarcomas in dogs, such as hemangiosarcoma, osteosarcoma, and histiocytic sarcoma.  In the veterinary oncology community, these tumors are not considered under the umbrella of soft tissue sarcomas given their very different biological behaviors.  Synovial cell sarcoma was also mentioned; this is a somewhat controversial diagnosis as there is likely much overlap with histiocytic sarcoma.  Even lymphangiosarcomas and rhabdomyosarcomas, while they probably best fit into this category, are commonly discussed separately. 

The most common histologic types in human patients (liposarcoma, leiomyosarcoma, and undifferentiated pleomorphic sarcoma) are exceedingly rare in veterinary patients, which makes the discussion of markers between the two a large reach if not a moot point entirely.  

Soft tissue sarcomas are listed in the manuscript as rare in both humans and dogs, but then the statistic of STSs representing 20% of all cutaneous tumors is listed.  Again, in the veterinary oncology community this is not considered rare. 

I believe the manuscript should be entirely reworked in order to be made more relevant.  If it continues to include a discussion of veterinary STSs, then much more discussion should be opened for the role of radiation therapy for treatment as well.  

Comments on the Quality of English Language

Only minor editing will be needed; the English language was fine.

Author Response

Thank you to the authors for submitting this work.  While it is very interesting and well researched for some aspects, my largest concern is that the discussion includes histologic types of tumors not typically classified as soft tissue sarcomas in dogs, such as hemangiosarcoma, osteosarcoma, and histiocytic sarcoma.  In the veterinary oncology community, these tumors are not considered under the umbrella of soft tissue sarcomas given their very different biological behaviors.  Synovial cell sarcoma was also mentioned; this is a somewhat controversial diagnosis as there is likely much overlap with histiocytic sarcoma.  Even lymphangiosarcomas and rhabdomyosarcomas, while they probably best fit into this category, are commonly discussed separately. 

A: We appreciated the suggestions provided by the reviewer since they largely contributed to improve our manuscript. To this aim we removed the term hemangiosarcoma, histiocytic sarcomas, synovial cell sarcoma, lymphangiosarcomas and rhabdomyosarcomas from the initial classification. Furthermore, we even removed from the text the section relating to canine hemangiosarcoma. Conversely, we did not remove the sections relating to rhabdomyosarcomas since they referred to human STS.

Given the paucity of information on canine STSs we used information relating to osteosarcoma to provide, by comparison, useful information relating to STSs. To this end, we have nevertheless stated the difference between osteosarcoma and STSs. Moreover, we have cited studies in the literature that highlight the usefulness of comparing the two diseases (line 261)

The most common histologic types in human patients (liposarcoma, leiomyosarcoma, and undifferentiated pleomorphic sarcoma) are exceedingly rare in veterinary patients, which makes the discussion of markers between the two a large reach if not a moot point entirely.  

A: We appreciate the reviewer's suggestion, but we would like to point out that, given the limited amount of information, the objective of the review is to give a preliminary description of the similarities between the two models. At the moment, it would be impossible to provide precise, timely and robust information given the bias introduced by a classification, in dogs, which is much less inclusive than the human one (Thway K. Pathology of soft tissue sarcomas. Clin Oncol (2009) 21:695–705. doi:10.1016/j.clon.2009.07.016; Canine Soft Tissue Sarcomas: Can Being a Dog’s Best Friend Help a Child? B. Seguin. Front. Oncol., 23 November 2017.Sec. Pediatric Oncology. Volume 7 - 2017 | https://doi.org/10.3389/fonc.2017.00285).

Soft tissue sarcomas are listed in the manuscript as rare in both humans and dogs, but then the statistic of STSs representing 20% of all cutaneous tumors is listed.  Again, in the veterinary oncology community this is not considered rare. 

A: We removed the attribute “rare” in relation to canine STS.

I believe the manuscript should be entirely reworked in order to be made more relevant.  If it continues to include a discussion of veterinary STSs, then much more discussion should be opened for the role of radiation therapy for treatment as well.  

Reviewer 3 Report

Comments and Suggestions for Authors

In this review paper, the authors aim to provide a comprehensive landscape, including the tumorigenesis, tumor microenvironment and its heterogeneity, and treatment options, of soft tissue sarcoma. The review is nicely written, covering many aspects with a clear focus on notable components such as fibroblasts and key genetic regulators in the context of STS. The motivation of this review paper is clear: the authors believe there are significant similarities between the soft tissue sarcoma (STS) landscape in dogs and humans. And it is from this angle that the authors provide an overview for both species to enlighten future studies on human STS, with references to findings in dogs. Using dog as a model for cancer research is not uncommon, but limited information is available in the context of STS. Therefore, this review is right on time and I recommend the publication. However, I do have a minor concern - please check the numbers in tables carefully, the available cases from ref.92 seems off to me. 

Author Response

In this review paper, the authors aim to provide a comprehensive landscape, including the tumorigenesis, tumor microenvironment and its heterogeneity, and treatment options, of soft tissue sarcoma. The review is nicely written, covering many aspects with a clear focus on notable components such as fibroblasts and key genetic regulators in the context of STS. The motivation of this review paper is clear: the authors believe there are significant similarities between the soft tissue sarcoma (STS) landscape in dogs and humans. And it is from this angle that the authors provide an overview for both species to enlighten future studies on human STS, with references to findings in dogs. Using dog as a model for cancer research is not uncommon, but limited information is available in the context of STS. Therefore, this review is right on time and I recommend the publication. However, I do have a minor concern - please check the numbers in tables carefully, the available cases from ref.92 seems off to me. 

A: We kindly thank the reviewer for the positive comment on the review submitted by us. As he suggested we also checked the bibliography.

Reviewer 4 Report

Comments and Suggestions for Authors

I thank the Authors for their huge work.

I think the manuscript is well written and structured. The topics included in their review make it complete and exhaustive.

I have not further suggestions

Author Response

I thank the Authors for their huge work.

I think the manuscript is well written and structured. The topics included in their review make it complete and exhaustive.

I have not further suggestions

A: We thank the reviewer for his kind words and hope that the work will be well received by the scientific community.

Round 2

Reviewer 1 Report

Comments and Suggestions for Authors

The authors replied to the review, “…we ask the reviewer, if possible, to indicate further works that can be cited in the text to better describe the background relating to STS in dogs and human counterparts. Indeed, we found a small number of papers on canine STS useful for a comparative analysis with humans.” It is not the reviewer’s job to help write the manuscript. If there are not enough published papers to be informative, then perhaps the authors ought to consider increasing the scope of the paper to include other sarcomas, such as osteosarcoma, where they may find more material. 

The paper is somewhat improved grammatically but unfortunately is still filled with errors. A number of those appear to have been added since the last review, indicating that English grammar review was not effectively conducted. While my concern about non-descriptive terms was addressed by removing some of those terms, my request that they be replaced with descriptive lists was not addressed. In addition, the authors use excessively verbose language on multiple occasions, peppering the manuscript with adjectives and adverbs that a generally not necessary to convey their point. 

Specifici issues with this manuscript include:

Throughout the paper dog and canine are used in inappropriate ways. Dog is a noun. Canine is an adjective. Phrases like “ to treat dogs’ STS…” would be better phrased “to treat canine STS”.

Line 50 – define pNST (Or PNST as it was previously written)

Line 89-90 – do 85% of subjects exposed to pesticides get STS? Or is it an 85% increase in STS in subjects exposed to pesticides?

Line 94-95 – The authors state “However, as reported by Edwards and colleagues, 94 onset of STSs onset is affected by other confounders as sex and age [19].” In their reply to the review, where I asked for more details, they responded, “The effect of age and sex cannot be known. The authors [of reference 19] themselves report the following sentence: “Many studies excluded female workers due to small numbers and rarity of involvement in highly exposed occupations. Therefore, subgroup analysis based on gender was not performed due to population selection bias.”” As a consequence, the statement in the current manuscript is not supported by the reference. If the effect of sex and age have not been analyzed, the authors cannot say that they affect STS onset.

Line 246-247 – The authors state “influences the prognosis…” but they don’t say if the influence is positive or negative. Does it improve the prognosis, or worsen it?

Line 260-261 – While osteosarcomas are common on the extremities, they are rare on the trunk and retroperitoneum. This sentence needs to be changed to reflect that.

Line 261-262 – “none the less” tends to mean “in spite of”. Here the next statement agrees with the preceding one and so “none the less” is not really appropriate.

Line 283 – “two clusters” – it is unclear what these clusters are. Clusters of dogs? of genes? Please clarify.

Table 1 is confusing. Why are the column titles repeated? The description under “Treatment Specification” don’t sound like treatments. The word “Exportation” seems inappropriate. This table needs significant revision to present whatever it is that the authors think it should be telling the reader (right now, I am hard-pressed to explain the purpose of this set of tables)

Line 375-376 – I am not aware that histology can predict chemosensitivity which appears to be the implication of this sentence. 

Line 380 – did the data “suggest” or did is show that reduction?

Line 387-388 – how are “histology and grade as well as tumor location”  fundamental “to provide an effective human STSs therapy.”? None of these provide therapy. They may be useful tools to decide which therapy may be most effective, but I don’t believe that they can predict response to chemotherapy.

Lines 390-391 – The authors write “…suggest that patients with high grade extremity STS measuring >10 cm may benefit…” Did their results show that it helped or not? The words “suggest” and “may” do not inspire confidence that the results being reported were significant. 

Line 396 – How is this the converse of what was said about humans? It appears that for both dogs and humans, radio- and chemotherapy are good options for larger tumors. 

Line 403 – the antibodies are to OX40. The current phraseology makes it seem as if the antibodies are called OX40, which they are not. In addition, it is not clear if there are also antibodies to TLR 3/9 or if some other form of agonist is being used. If it is not antibodies to TLR 3/9, then the nature of the agonist should be more fully explained.

Line 415-416 – if it is controversial, you should describe the two different outcomes, how they differed and any hypotheses on why they differed.  

Line 435 – 438 are written as if they summarized the previous section, but the conclusions presented have nothing to do with what has been presented to this point in this section. 

Section 3.1 describes immunotherapy, but the subject has already been partially discussed in lines 398-413. Perhaps this ought to moved to section 3.1.

Line 471 – it’s not really the converse if there is limited to no data. 

Line 478 – again, it is not “conversely” if both examples work in the same way. If the level of PD-L2 was not significantly different, you cannot say that it rose. 

Lines 530-548 – This discussion makes it sound like the human and canine studies had very different outcomes yet that doesn’t really seem to be the case with what was presented. If it is, please do a better job of highlighting the differences in the preceding parts of this section.

Some examples of spelling and grammar issues (not meant to be a comprehensive list):

Line 61 - should probably be “…inactivation of tumor suppressor genes…”

Line 91 – should be “led to” instead of “leads to”

Line 105 – the authors say “Firstly, they are exposed to humans’ similar environmental risk factors,” This needs to be rewritten to “Firstly, they are exposed to similar environmental risk factors as their human companions,...”

Line 107 – should be “susceptibilities”

Line 118-119 – should be “collaborators”’

Line 175 – should be “growth”

Line 175 – “VEGF can be used as prognostic markers…” should be “VEGF can be used as a prognostic marker…”

Line 191 – should be “In detail,” but the phrase is unnecessary and could be deleted. 

Line 227 – delete the added “s”. It should be 4-fold, not 4-folds.

Line 241 – it should be “inhibits”

Line 251 – the word “anyway” is unnecessary and should be deleted.

Line 254 – it should be “differs” and “diseases”

Line 260 – the authors say “Although osteosarcomas are different tumors from STSs…”, this might be better worded “Although osteosarcomas are not considered STSs,…”

Line 269 – should read “…associated with canine and human STSs.”

Line 269, 270 – two separate uses of “prognosis” would more properly be “prognostic” 

Line 181, 273, 446 – “Nowadays” is perhaps a bit colloquial and chatty for this type of manuscript.

Line 274 – the added “the” is not needed and should be removed. 

Line 277 – “profiling allowed to identify in canine…” should read “profiling allowed several abnormalities located in the phosphatidylinositol-4,5-bisphosphate 3-278 kinase catalytic subunit alpha (PIK3CA) and neuroblastoma RAS viral oncogene homolog 279 (NRAS) pathways to be identified in canine hemangiosarcoma.”

Line 289 – should be “… a complete absence…” and “such data…” should be “this data suggests…”

Line 291 – should be “a recent work on tumor transcriptomes allowed the demonstration of crosstalk between RNA…”

Line 299 – should be “tumor” not “tumoral”.

Line 305-306 – “…were proposed as new possible diagnostic and/or prognostic.” Should be changed to “were proposed as possible diagnostic and/or prognostic markers.”

Line 307 – should be “…human osteosarcoma…”

Line 308-309 – The existing sentence does not make sense. 

Line 310-312 - The existing sentence does not make sense.

Line 318-319 – should be “pathways”

Line 326 – should be disease, not diseases

Line 333 – should be canine (canine is an adjective, dog is a noun)

Line 334 - should be “tools”

Line 336-350 – There are errors in every sentence in this paragraph.

Line 351 - Should be “Surgical” 

Line 462 – should be “inhibitor”

Line 466-468 – The existing sentence does not make sense.

Line 521 – “goodness” is not an appropriate word here. 

Line 525 – “..capable to reduce…” is not appropriate wording.

Line 527 – the original “post operatively” was correct. The edit to “post operation” is not correct. 

Comments on the Quality of English Language

The paper is somewhat improved grammatically but unfortunately is still filled with errors. A number of those appear to have been added since the last review, indicating that English grammar review was not effectively conducted. While my concern about non-descriptive terms was addressed by removing some of those terms, my request that they be replaced with descriptive lists was not addressed. In addition, the authors use excessively verbose language on multiple occasions, peppering the manuscript with adjectives and adverbs that are generally not necessary to convey their point. 

Author Response

The authors replied to the review, “…we ask the reviewer, if possible, to indicate further works that can be cited in the text to better describe the background relating to STS in dogs and human counterparts. Indeed, we found a small number of papers on canine STS useful for a comparative analysis with humans.” It is not the reviewer’s job to help write the manuscript. If there are not enough published papers to be informative, then perhaps the authors ought to consider increasing the scope of the paper to include other sarcomas, such as osteosarcoma, where they may find more material. 

A: We reported, to our knowledge, all the studies on canine STSs. We had provided more information, relating in some cases to other form of sarcoma, but we eventually removed it as suggested by other reviewers.

The paper is somewhat improved grammatically but unfortunately is still filled with errors. A number of those appear to have been added since the last review, indicating that English grammar review was not effectively conducted. While my concern about non-descriptive terms was addressed by removing some of those terms, my request that they be replaced with descriptive lists was not addressed. In addition, the authors use excessively verbose language on multiple occasions, peppering the manuscript with adjectives and adverbs that a generally not necessary to convey their point. 

A: The paper has been revised again by a native English speaker to improve manuscript grammar and reduce the verbose language. We now added descriptive terms/lists at line 55, 75-76, 99-100, 418, 434.

Specifici issues with this manuscript include:

Throughout the paper dog and canine are used in inappropriate ways. Dog is a noun. Canine is an adjective. Phrases like “ to treat dogs’ STS…” would be better phrased “to treat canine STS”.

Line 50 – define pNST (Or PNST as it was previously written)

A: Amended as required

Line 89-90 – do 85% of subjects exposed to pesticides get STS? Or is it an 85% increase in STS in subjects exposed to pesticides?

A: We appreciate the comment since the sentence was not clear. We meant 85% increased risk of devoloping sarcoma compared to control.

Line 94-95 – The authors state “However, as reported by Edwards and colleagues, 94 onset of STSs onset is affected by other confounders as sex and age [19].” In their reply to the review, where I asked for more details, they responded, “The effect of age and sex cannot be known. The authors [of reference 19] themselves report the following sentence: “Many studies excluded female workers due to small numbers and rarity of involvement in highly exposed occupations. Therefore, subgroup analysis based on gender was not performed due to population selection bias.”” As a consequence, the statement in the current manuscript is not supported by the reference. If the effect of sex and age have not been analyzed, the authors cannot say that they affect STS onset.

 A: We deleted the sentence.

Line 246-247 – The authors state “influences the prognosis…” but they don’t say if the influence is positive or negative. Does it improve the prognosis, or worsen it?

A: We amended the sentence. We specified that TILS are associated to a better prognosis.

Line 260-261 – While osteosarcomas are common on the extremities, they are rare on the trunk and retroperitoneum. This sentence needs to be changed to reflect that.

A: We amended the sentence as required

Line 261-262 – “none the less” tends to mean “in spite of”. Here the next statement agrees with the preceding one and so “none the less” is not really appropriate.

A: The reviewer is right. We amended as required.

Line 283 – “two clusters” – it is unclear what these clusters are. Clusters of dogs? of genes? Please clarify.

A: As suggested we clarified the sentence by adding the term “transcript”.

Table 1 is confusing. Why are the column titles repeated? The description under “Treatment Specification” don’t sound like treatments. The word “Exportation” seems inappropriate. This table needs significant revision to present whatever it is that the authors think it should be telling the reader (right now, I am hard-pressed to explain the purpose of this set of tables)

A: In the present table we aimed to show a comparative description of different kind of treatment in dog and human. Table has been formatted entirely.

Line 375-376 – I am not aware that histology can predict chemosensitivity which appears to be the implication of this sentence. 

A: We reported the NCCN guidelines.

Line 380 – did the data “suggest” or did is show that reduction?

A: The data “show”. We used the term suggest since it is not an information widely reported in literature.

Line 387-388 – how are “histology and grade as well as tumor location”  fundamental “to provide an effective human STSs therapy.”? None of these provide therapy. They may be useful tools to decide which therapy may be most effective, but I don’t believe that they can predict response to chemotherapy.

A: We amended the sentence as suggested (“is fundamental to establish which therapy may be the most effective to treat human STSs”)

Lines 390-391 – The authors write “…suggest that patients with high grade extremity STS measuring >10 cm may benefit…” Did their results show that it helped or not? The words “suggest” and “may” do not inspire confidence that the results being reported were significant. 

A: The sentence has been amended as follow: “However, the results reported by Gamboa et al., show that patients with high grade extremity STS measuring >10 cm obtain benefit of neoadjuvant chemotherapy.”

Line 396 – How is this the converse of what was said about humans? It appears that for both dogs and humans, radio- and chemotherapy are good options for larger tumors. 

A: The reviewer is right. We deleted “Conversely”.

Line 403 – the antibodies are to OX40. The current phraseology makes it seem as if the antibodies are called OX40, which they are not. In addition, it is not clear if there are also antibodies to TLR 3/9 or if some other form of agonist is being used. If it is not antibodies to TLR 3/9, then the nature of the agonist should be more fully explained.

A: The reviewer is right. We modified the sentence to avoid misleading information.

Line 415-416 – if it is controversial, you should describe the two different outcomes, how they differed and any hypotheses on why they differed.

A: We used the term controversial since the treatment based on doxorubicin alone did not show particular benefit for patient outcome. Conversely, treatment based on high intensity focused ultrasound (HIFU) and doxorubicin showed promising results in the majority of patient. However, we deleted the term controversial (line 397).

Line 435 – 438 are written as if they summarized the previous section, but the conclusions presented have nothing to do with what has been presented to this point in this section. 

A: We removed such sentence.

Section 3.1 describes immunotherapy, but the subject has already been partially discussed in lines 398-413. Perhaps this ought to moved to section 3.1.

A: In the previous section we related on chemotherapy and radiotherapy. We cited immunotherapy only when combined with the previous two approaches. The section 3.1 is entirely dedicated to immunotherapy and we think it would be more useful for the reader to keep the two sections separate

Line 471 – it’s not really the converse if there is limited to no data. 

A: As suggested by the reviewer we deleted the term “conversely”.

Line 478 – again, it is not “conversely” if both examples work in the same way. If the level of PD-L2 was not significantly different, you cannot say that it rose. 

 A: As suggested by the reviewer we deleted the term “conversely”.

Lines 530-548 – This discussion makes it sound like the human and canine studies had very different outcomes yet that doesn’t really seem to be the case with what was presented. If it is, please do a better job of highlighting the differences in the preceding parts of this section.

A: As suggested by the reviewer we deleted the sentence “The discrepancies in clinical outcomes between the two models are multifaceted, involving not only disparities in immunobiological understanding but also limitations in the availability of critical research tools “ which appeared to be misleading.

Some examples of spelling and grammar issues (not meant to be a comprehensive list):

Line 61 - should probably be “…inactivation of tumor suppressor genes…”

A: Amended accordingly.

Line 91 – should be “led to” instead of “leads to”

A: Sentence has been changed.

Line 105 – the authors say “Firstly, they are exposed to humans’ similar environmental risk factors,” This needs to be rewritten to “Firstly, they are exposed to similar environmental risk factors as their human companions,...”

A: Changed.

Line 107 – should be “susceptibilities”

A: Amended as required

Line 118-119 – should be “collaborators”’

 A: Amended as required

Line 175 – should be “growth”

 A: Amended as required

Line 175 – “VEGF can be used as prognostic markers…” should be “VEGF can be used as a prognostic marker…”

 A: Amended as required

Line 191 – should be “In detail,” but the phrase is unnecessary and could be deleted. 

 A: Amended as required

Line 227 – delete the added “s”. It should be 4-fold, not 4-folds.

 A: Amended as required

Line 241 – it should be “inhibits”

 A: Amended as required

Line 251 – the word “anyway” is unnecessary and should be deleted.

 A: Amended as required

Line 254 – it should be “differs” and “diseases”

 A: Amended as required

Line 260 – the authors say “Although osteosarcomas are different tumors from STSs…”, this might be better worded “Although osteosarcomas are not considered STSs,…”

 A: Amended as required

Line 269 – should read “…associated with canine and human STSs.”

 A: Amended as required

Line 269, 270 – two separate uses of “prognosis” would more properly be “prognostic” 

 A: Amended as required

Line 181, 273, 446 – “Nowadays” is perhaps a bit colloquial and chatty for this type of manuscript.

 A: Amended as required

Line 274 – the added “the” is not needed and should be removed. 

 A: Amended as required

Line 277 – “profiling allowed to identify in canine…” should read “profiling allowed several abnormalities located in the phosphatidylinositol-4,5-bisphosphate 3-278 kinase catalytic subunit alpha (PIK3CA) and neuroblastoma RAS viral oncogene homolog 279 (NRAS) pathways to be identified in canine hemangiosarcoma.”

 A: Amended as required

Line 289 – should be “… a complete absence…” and “such data…” should be “this data suggests…”

 A: Amended as required

Line 291 – should be “a recent work on tumor transcriptomes allowed the demonstration of crosstalk between RNA…”

 A: Amended as required

Line 299 – should be “tumor” not “tumoral”.

 A: Amended as required

Line 305-306 – “…were proposed as new possible diagnostic and/or prognostic.” Should be changed to “were proposed as possible diagnostic and/or prognostic markers.”

 A: Amended as required

Line 307 – should be “…human osteosarcoma…”

 A: Amended as required

Line 308-309 – The existing sentence does not make sense.

A: Sentence has been modified

Line 310-312 - The existing sentence does not make sense.

 A: Sentence has been modified

Line 318-319 – should be “pathways”

 A: Amended as required

Line 326 – should be disease, not diseases

 A: Amended as required

Line 333 – should be canine (canine is an adjective, dog is a noun)

 A: Amended as required

Line 334 - should be “tools”

 A: Amended as required

Line 336-350 – There are errors in every sentence in this paragraph.

 A: Paragraph had been corrected

Line 351 - Should be “Surgical” 

 A: Amended as required

Line 462 – should be “inhibitor”

 A: Amended as required

Line 466-468 – The existing sentence does not make sense.

A: Sentence has been modified

Line 521 – “goodness” is not an appropriate word here. 

 A: Amended as required

Line 525 – “..capable to reduce…” is not appropriate wording.

 A: Amended as required

Line 527 – the original “post operatively” was correct. The edit to “post operation” is not correct. 

 A: Amended as required

Comments on the Quality of English Language

The paper is somewhat improved grammatically but unfortunately is still filled with errors. A number of those appear to have been added since the last review, indicating that English grammar review was not effectively conducted. While my concern about non-descriptive terms was addressed by removing some of those terms, my request that they be replaced with descriptive lists was not addressed. In addition, the authors use excessively verbose language on multiple occasions, peppering the manuscript with adjectives and adverbs that are generally not necessary to convey their point. 

Reviewer 2 Report

Comments and Suggestions for Authors

Thank you to the authors for their revisions.  I believe the paper is acceptable for publishing after minor wording and grammatical changes (see attached).

Comments on the Quality of English Language

Minor wording and grammatical changes necessary.

Author Response

Thank you to the authors for their revisions.  I believe the paper is acceptable for publishing after minor wording and grammatical changes (see attached).

Minor wording and grammatical changes necessary.

Page 2, line 62: be consistent with writing out gene acronyms as you have in subsequent sentences (for example lines 64-65).

A: Amended as required

Page 3, line 126 (and throughout text): the acronym should be STS (singular) here as it refers to the topic; when referring to the tumors, it should be plural.

A: Amended as required

Page 3, line 138: Be consistent with use of acronyms throughout paper (vs. spelling out soft tissue sarcoma)

A: Amended as required

Page 4, line 159: I would change back to “tumor cell (singular)” Page 6, line

A: Amended as required

227: change back to “4-fold (singular)”

A: Amended as required

Page 6, line 241 : change to “turn (singular)”

A: Amended as required

Page 8, lines 307-308: Reword to “patients’ exosomes; the analysis observed 20 microRNA upregulated...”

A: Amended as required

Page 9, Table 1 a: change “exportation”to “excision” Page 1 1,

A: Amended as required

Table 1 d: change to “caninized”

A: Amended as required

Page 9-12, Table 1 : ensure formatting is consistent.

A: Amended as required

Page 12, line 369: reword to “...effective to decrease the needed surgical margins without compromising patient outcome”.

A: Amended as required

Page 12, lines 375-376: reword to “...since such tumors rarely show highly chemosensitive histologic profiles more commonly.            ”

A: Amended as required

Page 12, line 392: remove the newly added “a”

A: Amended as required

Page 13, line 416: remove the newly added "a”

A: Amended as required

Page 13, line 418: remove the newly added "the"

A: Amended as required

Page 13, line 428: remove the newly added "the”

A: Amended as required

Page 1 4, line 472: change“few”to little

A: Amended as required

Page 15, line 508: end sentence after“innate immune response”.

A: Amended as required

 Page 15, line 51 1 : change to viruses

A: Amended as required

Page 15, line 527: I would personally change wording back to “post-operatively”

A: Amended as required

Page 16, line 557: change to “humans”

A: Amended as required

Page 15, line 562: change to “STS”

A: Amended as required
